# A qualitative study of Chinese teacher's perceptions and practices of meritocracy

**Xiaoxue Sun**[1]*, **Lan Shi**[2]

**1** School of Education (Teachers College), Guangzhou University, Guangzhou, China, **2** Institute of Education, Tsinghua University, Beijing, China

* sunxiaoxue_sxx@163.com

## Abstract

China has a long history of meritocracy, but as social inequality grows, people are increasingly questioning whether hard work promises a better life, even as national policies and mainstream media spare no effort to promote meritocratic narratives. In response, how do people interpret their lives and act within the conflict context between social realities and political forces? On the basis of semi-structured interviews with teachers in different types of schools, this paper explores how teachers interpret meritocracy and navigate it in their teaching practices. The results indicate that teachers show a dual attitude toward meritocracy. On the one hand, they believe that effort and ability are crucial to occupational and educational success, yet on the other hand, they also acknowledge the influence of *guanxi* on employment and the noticeable educational disparities caused by family background. Teachers have different approaches to balancing meritocratic and nonmeritocratic factors in their teaching. Teachers who limit their responsibilities regarding student growth offer verbal advice. The majority of teachers guide students to focus on working to redress the gap derived from nonmeritocratic factors while also warning students not to place too much hope on agency. Teachers' practices inevitably contribute to social inequality. This paper underscores that in an environment lacking redistribution mechanisms, meritocracy for teachers is more of a pragmatic calculation than a belief.

## Introduction

Meritocracy suggests that with equal opportunities, everyone can compete for social resources on the basis of merit, particularly educational qualifications. For example, the national entrance exam (*Gaokao*) still provides a path for people's upward mobility and empowerment in current China. The question is, does education-based meritocracy truly promise a chance for a good life? A consensus on contemporary Chinese education is that increasing numbers of children from upper-class families have gained places at elite universities, even as China has experienced an expansion in higher education since 1999 [1]. Moreover, graduates of top universities earn substantial premiums, as the labor market has changed with the expansion of higher education [2], further reinforcing the advantages of family status. Even if students from disadvantaged backgrounds gain entrance to elite universities, they often find themselves lagging behind their upper-class peers in the labor market [3]. Disadvantaged Chinese youth may recognize that their chances for mobility are relatively limited, thus prompting the "lying flat

**Data availability statement:** All relevant data are within the manuscript.

**Funding:** The author(s) received no specific funding for this work.

**Competing interests:** The authors have declared that no competing interests exist.

movement" [4]. Chinese researchers view the movement as a cultural representation caused by class solidification, where working-class youth are unable to achieve upward mobility through hard work [5].

Nonetheless, both national policies and mainstream media spare no effort promoting meritocratic narratives [6] and enhancing people's perceptions that the success of education and occupation can be achieved through hard work. Does such meritocratic propaganda work? Studies suggest that a belief in meritocracy is likely to be especially appealing to people, particularly when merit is the only way to achieve success and upward mobility [7,8]. Given the developmentalist model, Chinese people may exhibit a strong belief in meritocracy. According to Xiang [9], China has made little progress in promoting economic equity over the past 20 years, thus leaving people to rely on themselves to improve their lives. In contrast, studies have shown that people are less willing to believe in meritocracy when exposed to inequality cues, such as corruption and *guanxi* [10].

How do people describe their lives and take action in a society where mainstream discourse increasingly diverges from social reality? What impact do their actions have on society? This paper takes teachers as examples and captures how people feel, calculate and struggle by exploring teachers' perceptions of meritocracy and how they navigate the effects of meritocratic and nonmeritocratic factors in teaching.

The study integrates these findings with a discussion of the social inequality in China. In the next section, we briefly introduce the history of meritocracy in China, especially the government's merit-oriented policies after the reform and opening up, and then present the research methods and findings, followed by a discussion of the results and conclusions.

## Meritocracy and education

Drawing from Yang's description of meritocracy and its debates, regarding concepts such as "traps", "unfulfilled promises" and "myths" [11,12], meritocracy generally refers to a norm for distributing general social goods or a social system [13]. From its philosophical roots, meritocracy is viewed as a distributive principle. As Littler stated [14],

A meritocratic social system cannot be divorced from an understanding of how it functions in connection with the contextual issues of economic and cultural redistribution and recognition (in terms of, for example, the roles assigned and enabled in terms of physical ability, gender or caste) and with how social success is demarcated and financially and culturally rewarded.

(Littler, 2017: 9)

The merit principle is a distributive rule of justice that prescribes that an individual's relative outcomes (e.g., pay, grade and promotion opportunities) should be allocated in proportion to his or her relative inputs, e.g., effort [15]. Aristotle [16], for example, noted that justice is attained when people receive what they deserve and are rewarded for their merits. With respect to just distribution, the concept of desert (referring to what is deserved) is a synonym for meritocracy and is directly cited in many studies of meritocracy [17]. As with desert, merit is a pre-institutional notion that refers to merit determined rules entitled to rewards rather than biased rules [18,19]. Additionally, merit focuses on the principles of aboutness and agent-relative restrictions [20]. Children cannot, for example, have access to elite universities on the basis of being from wealthy families. One can only be a qualified candidate because of outstanding ability and hard work.

For the formulations of distribution, a society or political system that distributes social goods on the basis of merit is a meritocracy. The ideal of meritocracy is to create a connection between social mobility and sociopolitical and economic outcomes by sorting individuals on the basis of merit, thus ranking them into different groups and adjusting the composition of

social classes accordingly [21,22]. By analyzing the political selection system in China, Liu (2023) argues that a meritocratic political system can encourage people to seek economic mobility within the existing framework [6], thereby enhancing regime stability. Through the deliberative democracy process, meritocracy can enhance social economic efficiency, enabling both parties in transactions to benefit, according to Bruni and Santori [23].

If, theoretically, people can be rewarded on the basis of merit, then how does meritocracy judge merit? As criticized by Bovens and Anchrit Wille (2017), a convenient indicator of merit in modern society is the length of formal education, as indicated by the highest diploma earned [24]. Markovits holds the same view, i.e., the core assumption of contemporary meritocracy is that abilities and skills are acquired through learning [25]. As Alan stated, education is the bedrock of meritocracy and acts as the arbiter of merit [26], an interpretation also known as education-based meritocracy [27]. The theory of education-based meritocracy incorporates three factors, namely, class origin, educational attainment, and class destination [28]. Education-based meritocracy supposes that the link between educational attainment and class destination is strengthened as social selection increasingly relies on achievement, especially formal education [29,30]. In turn, the connections between class origin and destination (ODs) are mediated through education, with direct links gradually diminishing. According to logic, social mobility will increase, and if it does not, it is due to legitimate merit-based reasons. Optimistic researchers assume a positive view of the role of education when people with mixed abilities are placed in appropriate positions in the labor market and are also confident regarding the prospects for economic efficiency and social justice when based on education-based selection [31].

The appeal of education-based meritocracy is obvious and rarely challenged. However, the question remains as to whether meritocracy has fulfilled its promise. As empirical research has confirmed, meritocracy legitimizes social inequality and impedes social equity [32,33]. Success in education and occupied elite occupations is not due to abilities but rather to the specific features of one's background that make success possible [34]. As Bourdieu and Khan argue, the merit or ability recognized by society is actually the fruit of long-term investments by families in cultural capital and the construct of the educational system [35,36]. Meritocracy, however, obscures the connection between merit and privilege and instills a belief in merit [37]. The belief in meritocracy leads to decreased concern about social inequality. Mijs [38], for example, believes that meritocracy in school is associated with decreased concern about social inequality and diminished support for policies designed to reduce such inequality. Private school graduates who gain a place at elite universities believe that intelligence and hard work rather than privilege or parental influence allow them to study at elite schools [39], which leads them to oppose any form of 'compensatory sponsorship' even if they acknowledge the social barriers some students face, such as class, culture and schooling inequality.

Another argument concerning meritocracy is that merit-based distribution endorses endless competition that negatively impacts all classes and forms a linear hierarchy of winners and losers [40]. While the upper class holds an incomparable competitive advantage, they suffer enormous psychological pressure [41]]. In contrast, the lower classes with insufficient resources pay an even higher price. For example, those from the lower classes who succeed in joining elite circles struggle to imitate the upper-class habits, floating in the elite class yet being distant from their families [42,43].

## Teachers' perceptions of and practices related to meritocracy

Schools develop students' knowledge and skills while also conveying values, norms, and beliefs [11]. The center of informal belief lies in the meritocracy, which means that schools reward students on the basis of merit and effort (grades) rather than group memberships [44]]. Meritocracy in schools not only affects students but also influences society by shaping attitudes

toward social and economic inequalities [15,45]. Research suggests that schools' contributions to meritocracy are closely linked to that of teachers [46], making their practices and beliefs key to understanding meritocracy and social inequity. Teachers' perceptions and teaching practices also convey the informal values of the school [47]. In observations of daily school life, practices and rituals, Sousa demonstrated how teachers' narratives of discipline, determination and concentration instill meritocratic belief in students [48]. Studies such as Steinberg and Krumer's [49] and Calarco et al.'s [50] further suggest that teachers imbue students with meritocratic discourse through attributional preferences for academic success or failure, framing the way in which students discuss and respond to social barriers and agency.

Teachers who embrace beliefs in meritocracy usually interpret limitations in educational achievement as the result of the flaws and deficiencies of students and their families [51]. An international study by Batruch et al. [44] revealed that teachers with meritocratic beliefs, rather than perceiving unfairness and being supportive of affirmative action policies, encourage students to work hard, thereby perpetuating and strengthening meritocratic beliefs within schools. In a similar vein, scholars have found that in interactions with underprivileged black students, teachers tend to shift the subjects from structural issues to the force of personalities, implying that great character is shown in the struggle with poverty.

Attributing failure to individuals, educators may struggle to teach effectively, engage with, and advocate for children living in poverty [52]. A longitudinal ethnographic study of elementary and middle school classrooms conducted by Calarco et al. [50] found that a belief in meritocracy leads teachers to view students' struggles with homework through a lens of agency, rewarding those who meet expectations through familial support and punishing those who have difficulty meeting expectations owing to a lack of family support. In light of the same considerations, teachers blame families for failing to equip their children with the necessary skills for educational achievement, for not prioritizing education, and for being less involved, thus considering those parents to possess inadequate educational competencies [53]. Research suggests that teachers draw a line between agency and the challenges faced by families and students by emphasizing deficiencies and suggesting that effort can result in academic success without addressing social structural issues, thus reinforcing the meritocracy [54].

Despite the general consensus that meritocratic beliefs are prevalent and strongly affirmed among teachers [55], studies indicate that teachers are aware both of the challenges their students face and that structural factors, such as gender, poverty, and race, impact student agency, thus preventing them from readily blaming students. Teachers who disagree with simplistic victim blaming and perceive meritocracy as a pragmatic reality find themselves compelled to accept meritocracy as the cause for the failure of certain students [56].

Additional factors, such as social context, school circumstances, personal experiences, race and economic status affect the belief in meritocracy [57]. During interviews with teachers and students at a rural middle school in China, Hansen noted that personal experiences regarding poverty and rural status that are specific to individuals have resulted in meritocracy beliefs and behaviors, leading teachers to praise individual solutions to social challenges and view students who do not overcome those challenges as irresponsible and selfish [58]. The social context and school environment also shape teachers' beliefs in meritocracy. For example, in East Asian societies with a long tradition of meritocracy, teachers rarely question the relationships among personal effort, education, and elite positions [37].

## Meritocracy in the Chinese context

Meritocracy is regarded as an approach to governance in China, and is embodied in the educational institutions and examinations [59]. The Chinese empire has associated educational

institutions and examinations with civil service positions, selecting meritorious individuals to serve in public office, thus benefiting the entire society [60,61]. In contemporary China, meritocracy serves as an alternative to income redistribution and plays a significant role in regime stability by expanding the opportunity for education and civil service examinations. Studies such as that of Chua argue that meritocracy is not only concerned with governance in China but also a major factor in the distribution of social goods following the increased economic growth due to China's opening-up policy [62].

Even though meritocracy has a long history in China, the history is complex. Meritocracy was suppressed in Maoist China, where political favoritism and egalitarianism were the exclusive standards [63]. Since the 1970s, the Communist Party of China (CPC) has introduced a series of policies with two points, open-door policies and efficiency over egalitarianism, which have signaled a return to meritocracy and highlighted education's screening function [64]. Hence, the CPC resumed the college entrance examination system (Gaokao) and then applied the meritocratic principle to political selection through the civil service examination in approximately 2000.

Throughout the subsequent decades, merit-oriented education was a basis for promoting social equity in the Chinese government. During President Hu's tenure, he made the famous statement that educational equity is the foundation of social equity [65]. Increasing investments in education, providing educational training, and guaranteeing educational opportunities to improve student quality have been common themes in the Party State's response to educational and social inequalities [66]. For example, President Xi believes that education is a critical path to improve the quality and skills of the impoverished population and end the intergenerational transmission of poverty [67]. Xi stated,

Poverty eradication must rely on intellectual support; combating poverty starts with combating ignorance. To ensure that a broad range of students grow healthily and happily and to provide every child with the opportunity to stand out, it is essential to broaden the channels for upward mobility for impoverished students and create more equitable access to education for students from poor regions (MOE, 2022).

Recently, as the government has sought to achieve greater equity through an economy driven by technology and innovation [68,69], it has focused on improving the quality of elementary education, such as increasing the professionalism of teachers, adopting scientific teaching methods, fostering multiple competencies of students, setting uniform standards in schools, and increasing the quality of higher education. First-rate universities and academic disciplines (Shuang yi liu) were introduced accordingly, as they are assumed to be vital to the goal of achieving equity, according to the MOE. As researchers have suggested, despite carrying the banner of equality and justice, education policies are not based on considerations of equal rights and welfare for citizens; instead, they legitimize the unequal distribution of resources and rights under the guise of merit [70].

Using the employment of college students as an example, the number of graduates in 2022 exceeded 10 million for the first time, according to China's largest online recruitment platform, Zhaopin. A total of 42.8% of graduates from four-year universities received job offers before graduation, as did 58.9% of graduates from elite universities (first-tier universities). Due to high unemployment in 2023, the government stopped publishing the unemployment rates among individuals under 24 years old starting in August of that year [71]. Not surprisingly, underprivileged students are more affected by the dim employment prospects. Moreover, rural students from disadvantaged backgrounds are prone to unemployment or underemployment and earn lower wages [72]. In contrast, students from families in urban areas (体制内) with abundant political resources and high incomes usually occupy upper-level jobs due to advantages in *guanxi* (关系) or receive valuable

information, which is also considered a primary reason for labor market inequality among graduates [73].

Moreover, the government has not implemented redistributive measures to address the disparities [74]. Instead, in addition to emphasizing the quality of education spurring economic growth, which in turn creates jobs [75], the party–state advocates social mobility through a meritocratic system, encouraging people to rely on efforts to alleviate their dissatisfaction with social inequalities. The recent expansion of graduate education and the National Civil Service Examination (NCSE) exemplifies this approach [76]. In 2022, the number of graduate students enrolled in China was 1.24 million, accounting for 20.9% of the total undergraduate and graduate enrollment that year (5.92 million). Moreover, compared with 2018, 6.6 million graduates registered for the NCSE in 2022, a 41% increase [77]. Among the graduates that year, 40% of the students from first-tier universities choose to pursue graduate studies, whereas nearly half of the graduates from four-year universities prepared for the civil service exam [78]. The competition for both graduate school entrance and civil service exams is highly intense. The acceptance ratio for a master's degree was 4.2:1, whereas the admissions rate for the 2021 NCSE was 1:61 [78]]. Nonetheless, a general belief among candidates is that the recruitment system provides equal opportunities for all graduates, and everyone can succeed on their own [73]. Accordingly, those deemed "successful" are considered to possess merits and social values that differentiate them from those who are not as successful.

The paradox of meritocracy, as Mijs argues, is that despite growing social inequality, the belief that disparity results from ability is becoming even more popular. As seen in China today, the changing economic climate, policy incentives, and welfare system have all contributed to the myth of meritocracy and have had a real impact. A study of how the Chinese perceive inequality revealed that the Chinese have a surprisingly strong belief in meritocracy, which results in a social volcano of inequality leading to conflict that has remained dormant [79].

## Method

### Participants

The participants were 24 teachers from three high schools in Shenzhen City, Zhejiang, and Henan provinces. Studies suggest that teachers in schools with different socioeconomic statuses (SES) vary in terms of their teaching practices [80]. To increase the diversity of perspectives, we sought participants from both upper-SES and lower-SES schools. Studies have also confirmed that socioeconomic background predicts meritocratic values [11]. Concerning the association between geographical location and income welfare benefits in contemporary China [81], participants came from public schools in distinct economic regions. We also focused on the effects of personal characteristics, such as educational level, gender, teaching experience and position, on teachers' views and practices. After combining the three criteria while considering availability for interviews, 24 teachers were sampled from the potential participants.

As Table 1 shows, seven came from School A, a key public high school located in Shenzhen, one of China's most developed cities. These seven teachers were not Shenzhen natives; rather, their advanced degrees, particularly as they were from elite universities, made them highly sought-after candidates for recruitment in flagship high schools like School A, and they immigrated to Shenzhen city. It should be noted that China's university system operates in distinct tiers reflecting institutional prestige, resources, and selectivity. Elite universities are the 985 Project (launched in 1998 to build China's "Ivy League") universities,

**Table 1. List of respondents.**

| | School with upper SES | School with upper-middle SES | School with lower SES |
|---|---|---|---|
| **Gender composition** | 2 male/5 female | 5 male/4 female | 4 male/4 female |
| **Median age** | 34 years old | 40 years old | 44.5 years old |
| **Institutional Prestige** | 6 (elite universities)/ 1 (1st-tier)[a] | 2 (1st-tier universities)/ 7 (2nd-tier)[b] | 5 (2nd-tier universities)/ 3 (higher vocational colleges)[c] |
| **Geographical background** | 6 (new immigrants)/ 1 local | 2 (new immigrants)/ 7 locals | 0 (new immigrants)/ 8 locals |
| **Master teachers (backbone teachers in Chinese)** | 2 | 1 | 0 |
| **Graduate-level or higher degrees** | 7 | 4 | 2 |
| **Total** | 7 | 9 | 8 |

[a]Teachers from Project 985/211 universities.

[b]Teachers from locally prestigious universities.

[c]Teachers from vocational colleges.

which receive significant government funding and enjoy the highest prestige. Of the 1,239 comprehensive public universities in mainland China, there are only 39 such institutions (MOE, 2022). First-tier universities refer to 211 Project universities (72 in total, excluding the 985 Project universities) and institutions with specific top-ranked programs (32 in total, excluding 985 and 211 institutions; MOE, 2022). These universities are able to receive partial government funding and enjoy regional reputations. Second-tier universities are regular higher education institutions that are mainly funded by local governments, have lower entry requirements than elite and first-class universities, and emphasize practical skills and undergraduate education over research. Higher vocational colleges offer 2–3-year programs focusing on employable skills and are seen as an alternatives to workforce-oriented academic degrees. School A prided itself on its highly qualified faculty, with approximately 30% of teaching staff holding mid-to-senior level academic positions [82]. Two of the seven teachers interviewed possessed senior-rank professional certifications. Nine came from School B, a highly selective urban high school in a moderately developed province known for its intense competition in college entrance examinations. Postgraduate degree holders constituted 18% of School B's teaching staff, compared to 60% at School A [82]. School B's faculty composition showed provincial characteristics, with approximately 90% of academic staff comprising locally trained educators who had completed their initial teacher training at regional pedagogical institutions [83]. Seven of the teachers interviewed at School B were from the local city or its subordinate counties; four had master's degrees, whereas the remaining teachers had bachelor's degrees.

Eight teachers were from School C, which served left-behind children (LBCs) in an impoverished county whose parents included migrant workers, manual workers, and street vendors. The majority of School C's teaching staff were from county-level jurisdictions and affiliated townships, with over 15 years of teaching experience. Their initial qualifications consisted of associate degrees, which were upgraded to bachelor's degrees through in-service continuing education programs [84]. Recent years had seen the school replenish its faculty through China's Special Post Teacher Program [84], a government-sponsored initiative addressing rural education gaps. Among the interviewed faculty members, two were under age 30 and held graduate-level degrees.

The teachers, who included 11 males and 13 females, had between 5 and 33 years of teaching experience. Thirteen were head teachers, and the remainder were subject teachers.

### Interview outline

We used semistructured interviews to explore teachers' perspectives on meritocratic and nonmeritocratic elements with respect to educational and occupational success. Drawing on the research literature, Sun drafted an outline for the interviews. To identify any questions that needed to be clearer or more straightforward for the participants, we conducted field interviews with five teachers who had submitted consent forms. Questions were revised accordingly. These 5 teachers did not participate in the subsequent formal interviews. In accordance with the participants' responses on the field test, we added questions regarding the teachers' views on intelligence factors in formal interviews and removed the ambiguous question about how teachers defined success (see S1 Appendix).

### Procedure

Approval for the study was received from the Research Ethics Committee of Guangzhou University ([2023]077), and the data were collected through one-on-one semi-structured interviews. The participants were recruited through the Chinese social application WeChat between June to July 1, 2023, and one hundred and seven potential participants expressed their willingness to participate in the study and passed the basic qualification screening. Researchers conducted purposive sampling of the potential participants on the basis of gender, teaching experience, teaching subjects, living region, type of school, and level of education. After the participants were confirmed, we mailed them a hard-copy informed consent form. The participants signed the form, scanned it, and emailed it back to us. We received consent forms from 30 teachers. We also obtained verbal consent from the participants at the beginning of the formal interviews.

Before the interview, the three researchers held a working meeting to standardize the data collection procedures. The meeting clarified core questions, follow-up questioning strategies, and question sequencing while establishing boundaries for the researchers' behavior to minimize potential biases. We also defined a specific method for observing and consistently documenting nonverbal cues. During the interviews, the researchers were asked to obtain participants' consent for full audio recording and to collect nonverbal information. A mid-process group meeting enabled the researchers to share their experiences, allowing us to address potential inconsistencies or biases. Post-interview procedures included sharing and reviewing the audio recordings to ensure consistency and comparability of the data collected by different interviewers. Following Becker [85], we stopped adding cases when we no longer received any opposing data, and we reached data saturation after interviewing 24 teachers. The interviews, each of which lasted approximately 60 minutes, were separately conducted in Mandarin through video conferences by three experienced qualitative researchers between July 10 and September 1, 2023. Each participant received a RMB 100 payment (wei xin hong bao) or equivalent gift as compensation. All participants' names have been anonymized.

### Analysis

The data were subjected to thematic analysis focused on the entire research phenomenon [86]. While guided by the research questions on 'teachers' perceptions of meritocracy' and 'teaching practices', the authors read through three interview transcripts to develop a brief coding guide containing operational definitions for meritocracy beliefs and questions (including code names, definitions, typical examples, and exclusion criteria). Before the formal coding, two authors conducted the calibration exercise and independently coded 10% of randomly selected transcripts [87]. Inter-coder reliability was assessed using Holsti's formula, agreement was high (.93) for thematic codes and good (.89) for character count. Coding disagreements

were mainly in the judgment of non-merit factor boundaries. Therefore, we included more raw data examples and designated *guanxi* as an analytical category, given it was frequently quoted in teachers' perceptions of occupational success and its critical role of *guanxi* in Chinese society. We refined the coding framework and confirmed three categories: (1) efforts, (2) guanxi, and (3) teaching practices. To validate the coding framework, conflicting codes were documented and reviewed through negative case analysis (e.g., narratives in which meritocracy and *guanxi* coexisted).

The two authors conducted independent formal coding, documenting the entire procedure including rule adjustments and emerging queries. They subsequently reviewed the coded results through consensus meetings, grouping the types of disagreements. They discussed coding disagreements on a line-by-line basis, with a focus on verifying: whether the boundaries of the code guide had been breached; and whether there were emerging themes. Through discussions with a third researcher involved in data collection, we finally reached an agreement (e.g., the conceptual classification of intelligence as an ability) via an arbitration process for contentious cases. In the end, we conducted member checking by presenting the finalized coding framework to a randomly selected teacher participant, confirming that the thematic interpretations aligned with their experiential narratives

## Results

### Merit-based formula for educational success

The participants use meritocratic factors to explain students' academic success or failure. Views are found among teachers from highly selective B schools, who advocate education-based meritocracy as the overall dominant approach to success. Accordingly, teachers take great care to inculcate the values of individual agency on college entrance exams and status chasing. When students fail, teachers persuade them to turn inward, examine themselves, and redouble their efforts to obtain another chance. Teacher W in School C comes from a rural area in a county where the per capita net income is below the poverty line of a monthly income of 150 CNY per capita (Central People's Government, 2013). Through her efforts, she attended a four-year university and obtained a teaching position after graduation by passing a public examination. She accounts for some of the students' failures in the following ways:

They lack self-discipline and are very susceptible to the learning environment and family circumstances. Most are left-behind children whose parents are usually absent. Some students are adrift, have no goals, have low motivation, and are empty-headed... It is a matter of learning habits formed in elementary and middle school, an inertia that manifests as a lack of initiative and a reluctance toward hardship. (C, 20220811)

Although teachers favor meritocracy, there is still some variation. Mr. H from School A is a youthful educator teaching geography with a full-time doctoral degree and only two years of teaching experience. He commented on his students as follows:

They do not resort to studying or working hard to change their fate, as most students in China do. They just come to school. Regardless of what they study in college when they graduate, they either inherit the family fortune or join the family business or company. (A, 20220925)

He also acknowledges the value of effort and merit. Additionally, he does not deny that families' ample resources map out more inclusive strategies for the success of children who do not rely on the education system. In addition to effort, it is noted that teachers at School A prioritize talent that requires time, cultural capital, and family background, according to Bourdieu [35]. They explain excellent academic performance by emphasizing the intelligence quotient (IQ). IQ, which is difficult to measure without controversy, is approximated by more

observable qualities, such as the ability to use inductive reasoning and quickly grasp new concepts. Yang, a natural science instructor at School A, confirms that almost all elementary education teachers acknowledge the importance of intelligence. They believe that the gap created by talent is reasonable and cannot be overcome. Not only teachers but also students endorse talent. Mr. Su from School B quotes a phrase circulating among students in his class:

Learning follows a three-to-seven ratio—three parts innate ability to seven parts diligence. It can be frustrating. Sometimes, no matter how hard you try, the pieces just won't fall into place. (B, 20220905).

Even with effort constituting the smaller portion of academic success for students, they still deride underperformers as engaging in mere "performative diligence." The students believe that losers pretend to work hard simply to comfort themselves. Alternatively, one can never surpass innate talent with mere effort alone.

While teachers at the other two schools recognize the significance of intelligence, they also believe that instilling hope in those struggling with their studies is important. Therefore, to manage course difficulties and engage in learning, they stress to their students that attitude and devotion to learning contribute to success. A teacher in School B states,

Even if you fail to grasp the subject, as long as you are willing to learn and have a positive attitude, it will eventually benefit your future work and life. (B, 20220901)

In addition, the respondents find it difficult to disregard the debate on the dynamics of education–family backgrounds, the core of which is the "reproduction of advantage" and the challenges of meritocracy. While the educators are reluctant to connect students' struggles with their social class fate, they insist that converting family capital into capability does not exclude determined efforts and that uncertainty in the conversion process offers an alternative method for vulnerable students. Even so, there is consensus among the participants (22 of 24) that family circumstances, such as economic status, especially mothers' free time, parental education attainment, and social capital, play crucial roles in converting students' effort and talent into attainments in education and occupation.

A teacher in School B believes that school is often less efficient than family. Even though the typical features of high school education, such as standardized curricula, intense competition, and the popularity of shadow education, may suppress the potential benefits of social origin [88], privileged parents always know how to align, or even increase, the tempo of school life. Accordingly, the teachers accept that there is a deficient family behind most failing students. Most School C teachers tend to view families with low socioeconomic status and undereducated parents as deficient. A teacher from School C discussed family challenges:

Why do excellent kids keep getting better? Because their parents see their potential and are willing to invest in it. They provide continuous support to ensure their progress and stability. Some students, however, struggle academically from a very young age because they have not developed good habits at home. They might think, "If worse comes to worst, I will just go out and work like my parents." They lack models and cannot experience the benefits of learning from their parents, so they do not care about education as their parents do. They might even think that working like their parents is a good-enough option for their future. (C,20220812)

## Who do you know? Effects of *guanxi* on occupational attainment

Most of the participants (21 of 24) agree that nonmeritocratic elements are as important as meritocratic factors in job seeking. The qualities emphasized in the school setting, such as taking responsibility, resilience, and self-discipline, are still acknowledged in the workplace but no longer endow the owners with exclusive rewards. Moreover, the criteria and principles for the most prestigious positions are imprecise and flexible, especially with respect to positions

in government public services [30]. The point is that elites or sponsors appreciate the individual abilities and talents that are paramount for social climbers. Teachers generally embrace the discourse that qualifications are just a steppingstone opening the door to opportunities, but the real challenge lies in establishing and leveraging *guanxi* behind the door. In the portraits of the elite created by teachers, competence, having *guanxi*, and understanding social norms are further evidence of combined beliefs in meritocratic and nonmeritocratic elements. Consequently, when a personal relationship is used, an ambitious climber can convert performance and talent into practical opportunities. A teacher at School B, when we asked what it takes to achieve occupational success, stated,

Ability and sponsors. You must have skills; if you possess what no one appreciates, you will not get far... It could be a word for leaders: you are adequate, and I will accept you. So, it would be best to strive to make big men know you, like you, and acknowledge your abilities. Otherwise, your skills, talents, and efforts may not be exploited even if you are capable, and these two are consistent. On the other hand, even if they know you, you also need to be qualified. What is the point of them knowing you if you cannot do your best at your job and you are ignorant of *renqing shi gu*? (B, 20220904)

The term '*renqing shigu*' (人情世故) literally means "the feeling of relationships, especially in family life [89]. In this sense, *renqing* primarily refers to adhering to a ritual order that emphasizes hierarchy based on age, status, gender segregation, appropriate acts and the emotions arising from such interactions [90]. The emotional dimension of renqing has undergone a shift in communist China, with an emphasis on human obligations and favors [91,92]. In Blau's classic interpretation of obligations, when people provide goods and services, it obligates the recipients to reciprocate [93]. Obligations in *renqing* embody reciprocity—people may alternate roles as providers and receivers of benefits [94]. King and Young's research argues that reciprocity in *renqing* is the basis of *guanxi* [95]. Ways to build *guanxi* include appropriate acts that align with the morals or norms of *renqing*, strengthening a bond of commitment, and creating an attachment between individuals through reciprocity [96]. In this context, *renqing* serves to maintain *guanxi* by fulfilling two functions: it outlines behavioral codes compatible with the social interactions that constitute *guanxi*, while also providing emotional validation that reinforces personal commitments to *guanxi* practices [97].

As such, the "emotional quotient" (EQ), a variable related to *renqing* frequently mentioned by teachers, provides basic scaffolding for constructing specious versions. Students have been taught to nurture *renqing*, prompting appropriate interpersonal relationships with others within or across groups, especially those ranking high in the social hierarchy. The teachers' refusal to idealize individual agency is demonstrated by their noncommittal responses when they are asked whether high achievers who are not familiar with *renqing* will achieve success in their careers. Similarly, in favor of high achievers, the participants did not think that high achievers were better than were studious people. Two teachers from schools B and C said,

Children from well-off families usually have high emotional intelligence; they are polite, know how to respect their teachers, and can sometimes be very close to you. Interacting with them can often be quite pleasant. (B, 20220909)

Some students are only good at studying and dedicated to academics. They do not socialize with others and lack social skills, so they may not do well in the workplace. (C, 20220811)

The teachers suggested that social origin and merit are mutually reinforced in the workplace. These values can also apply to academic achievement, yet how they do so differs. Within the education system, family privilege is embodied in disposition and justified by the Gaokao, facilitating social reproduction. Hence, a meritocratic explanation is satisfactory. In contrast, the occupational structure provides room for social resources to transform. High-status

parents activate diverse capital to generate advantages for their children within various social systems. Interestingly, teachers refer to the correspondence between students' trajectories in the labor market and family resources as rules of the real world. These bridging ties also affect how the teachers navigate meritocracy in their teaching [98].

## Be better than yourself, not others: coordinating the meritocratic and nonmeritocratic factors in teaching

With respect to how students remain unaffected by disadvantaged home environments to bridge the gap with others, the participating teachers are willing to help them. However, teachers do not share the same practices. Those who see themselves as servers have limited responsibility for the challenges students face; thus, their assistance depends on the efforts that students and their families have made. Typically, they give only verbal advice or adopt a detached attitude toward student development rather than becoming actively involved in promoting change. In commenting on how to meet the needs of low-status students, Hui, a teacher, is an experienced yet disillusioned educator who places great emphasis on the role of the family and holds little hope for the impact of his work. He stated,

Peers, family, and society shape students' development; the role of teachers is minimal. I would consider his academic performance and family situation to give him some suggestions. That is probably where my involvement would end. (A,20220922)

When a situation cannot be changed, one can ignore it and play the "game" in a different way. The typical approach that most teachers take is to avoid discussing the tension between individual agency and social exclusion in their teaching. By constantly emphasizing that 'education changes one's fate,' teachers believe that students can make a difference by trying their best. Even if nothing changes in the end, at least morally, they cannot be faulted for the approach. The meritocratic logic is transplanted into unpredictable job markets. To educate students to adopt and leverage *guanxi* for their benefit, teachers appoint class cadres to create interpersonal opportunities in the hope that they can learn how to handle relationships, especially with teachers, the authority in the organization. This practice may help high-achieving students be judged favorably by their superiors and be given sponsored mobility [99]. For poor performers, most teachers convey that this arrangement may compensate for their cognitive skills.

In addition to helping students improve their abilities, teachers guide them to find suitable paths. We must acknowledge that the phrase "suitable for her" does not merely involve a matter of choice that presents how educators redefine "success" and allow students to visualize it but also masks inequalities in educational gains through meritocracy. A teacher in School C noted,

He is inferior. He is deficient compared with his peers or those from privileged families, but his efforts are for his own sake, ultimately to make something of his life. So, trying to catch up with or surpass others is indeed impossible due to the home environment and social class. But you need to know that you are living your own life. You will eventually go into the real world and have a family, and your efforts are to improve your future. You just need to compare you with yourself, not necessarily outdo others. (C,20220826)

The terms "compare you with yourself" and "suitable for you" are often cited by teachers from School C to encourage students. However, theose phrases suggest a double standard for success because someone is destined to fail. These students must also respect the rules of the game, lower their exceptions, and compete on their own terms for success. According to Bourdieu [100], lower expectations adapt the economically poorest individuals to the specific conditions in which they find themselves.

## Discussion and conclusions

This study examines teachers' views of meritocracy and how they balance structural constraints with agency in teaching practices. In general, teachers' belief in meritocracy is flexible (see Fig 1). As noted, the interviewees widely believed in meritocracy, even while acknowledging significant educational disparities caused by family background. This phenomenon can be explained by the reality that, while an established practice in families involves investing in children's human capital—through cultivating skills and attitudes valued by society, monitoring and intervening in their education, purchasing supplementary educational services, and creating safety nets if educational plans fail—these advantages cannot be directly inherited. As a result of intense competition for admission to elite universities and the prioritization of standardized examinations in China's education system to assess curriculum-based learning ability, compels students from upper-class families to meet schools' relatively neutral academic criteria and demonstrate comparable academic performance to their disadvantaged peers, even though they are still more likely to access prestigious institutions [101]. The intense competition requires tremendous effort from all participants, regardless of socioeconomic status. It is this observable correlation between effort and academic success—exemplified by exhausted students entering top universities—that reinforces teachers' confidence in the role of individuals' agency.

Teachers adopt a cautious attitude toward meritocracy in the context of occupational success. This skepticism becomes pronounced where there is heavy reliance on *guanxi* in job-seeking, as the implicit advantages derived from *guanxi* undermine meritocracy. The influence of *guanxi* on labor market returns is expanding amid China's ongoing social transformation, a process characterized by unclear institutional rules, inconsistent enforcement, and mutually incompatible policy objectives [102]. The unique competitive edge provided

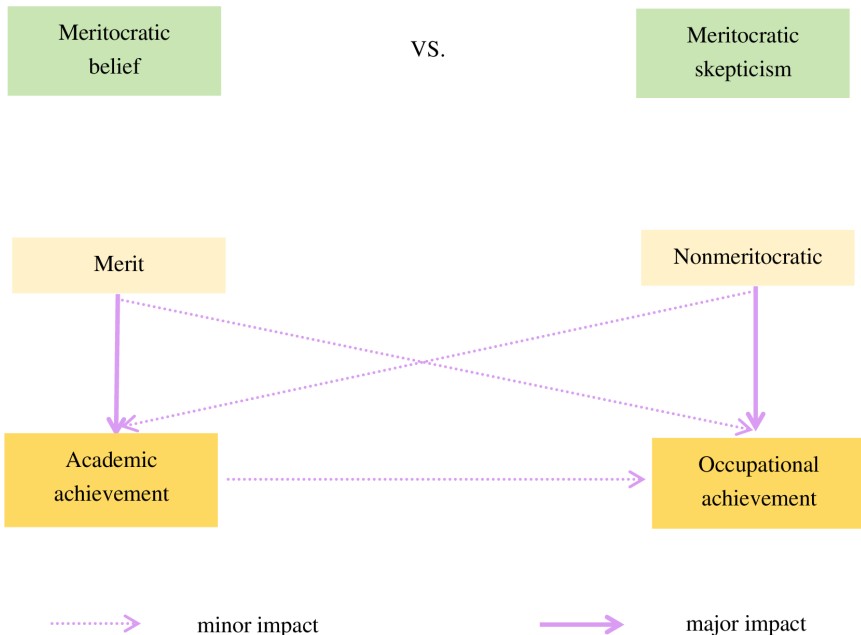

**Fig 1. Teachers' assumptions regarding the impact of merit and nonmerit on achievement in school and occupation.** This figure displays the perceived contributions of merit-based factors (talent vs. effort) and nonmerit factors (family socioeconomic status vs. *guanxi*) to students' educational achievement, with a direct comparison to their impact on occupational success.

by *guanxi* (its inherent value, limited availability, and inimitability) makes ability and effort insufficient for success. Consequently, students' transition from education to employment increasingly depends on family resources rather than merit.

Despite the limited rewards of effort in the labor market, teachers continue conveying meritocratic narratives to students. This tension between belief in and doubt regarding meritocracy runs through educational practice. On one hand, teachers are willing to reward diligent students even when their contributions have not yielded corresponding returns and urge economically disadvantaged students to bridge achievement gaps through hard work. On the other hand, they temper students' expectations of the rewards, highlighting effort as a path to moral fulfillment rather than socioeconomic improvement. This contradiction between beliefs and teaching practices reflects educators' role strain—where emphasizing effort serves both as professional responsibility fulfillment and psychological self-preservation amid structural constraints beyond their control—which inevitably makes teachers agents perpetuating social inequity, a critique prevalent among researchers [103]. We propose reframing teachers' ambivalence through the lens of national developmental challenges, not merely as acceptance of unequal systems.

As Harvey argues, teachers who depend on the system and lack power choose to support meritocracy rather than the meritocratic narrative that shapes how they discuss difficulties [80]. Chinese teachers act within constrained autonomy in both educational reforms and the classroom. Educational reforms usually follow a top-down model: the Ministry of Education (MOE) initiates policies that regional education departments establish plans to implement. Schools and teachers merely enforce policies, documenting the enforcement for higher authorities to assess and monitor adherence. This centralized pattern emphasizes control and obedience in educational management, marginalizing teachers from school decision-making [18,104]. While the MOE encourages teacher innovation in education reform, such efforts often lack institutional support. In particular, there is no formal organization for critical discussion of curricular design and knowledge systems, a space that could enable real change [58]. At the school level, performance evaluation mechanisms intensify pressure through a measurable index assessing teaching quality, professional ethics, competency. As performance directly determines financial and symbolic rewards or punishments and severe consequences (income reductions, being sidelined and diminished promotion prospects) for deviation, teachers are forced to comply with school demands and struggle to explore alternative teaching approaches [105]. As noted by a teacher at School C: "Ordinary people are on the edge of an abyss without relentless effort." Rather than conveying optimism about agency, this statement describes a desperate struggle with powerlessness.

Accordingly, calling for teachers or schools to address issues rooted in social structures and institutional frameworks is more like the educationalization of social problems [106]. To protect vulnerable groups and advance equity, the Chinese government need reform the current benefit and welfare distribution structure and uneven social protections. This is understandably complex and difficult, given that the party-state still retains control of high-revenue sectors. In this sense, education-focused policy interventions face inherent limitations in scope and impact. Nevertheless, as Yang Hongzhi and Clarke contend, even in these constrained contexts, engaging teachers in research and reflective teaching can develop agency and generate unexpected outcomes. To make this happen, educators can attempt to take the first step toward positive change by educating themselves not only about how privilege operates in daily life but also how it can be mediated by formal organizations or informal social networks. By doing so, teachers can move toward teaching with empathy and understanding, fostering a more equitable and inclusive educational environment.

From teachers' perspective, the contribution of this paper is to examine how schools shape students' meritocratic beliefs from the teachers' perspective and legitimize social inequalities. Given that meritocratic beliefs can be passed down intergenerationally, future research should consider a family perspective to examine the role of schools and families in shaping students' values [107]. Another limitation of the analysis is that it is exploratory and does not provide causal evidence for the potential legitimizing effect of teachers' behaviors on inequality. Future research should consider the collection of more comprehensive data to validate these correlations. In conclusion, this study provides a new perspective on meritocracy in contemporary China and provides a pathway for understanding how social inequality is legitimized through meritocratic beliefs.

## Supporting information

**S1 Appendix. Interview questions.**
(DOCX)

## Author contributions

**Conceptualization:** Xiaoxue Sun, Lan Shi.

**Data curation:** Xiaoxue Sun.

**Formal analysis:** Xiaoxue Sun, Lan Shi.

**Investigation:** Lan Shi.

**Methodology:** Xiaoxue Sun.

**Supervision:** Xiaoxue Sun.

**Writing – original draft:** Xiaoxue Sun.

**Writing – review & editing:** Xiaoxue Sun, Lan Shi.

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
