## [Decision Letter · Decision Letter 0]

18 Oct 2024

PONE-D-24-28202A Qualitative Study about Chinese Teacher’s Perceptions and Practice of MeritocracyPLOS ONE

Dear Dr. Sun,

Thank you for submitting your manuscript to PLOS ONE. After careful consideration, we feel that it has merit but does not fully meet PLOS ONE’s publication criteria as it currently stands. Therefore, we invite you to submit a revised version of the manuscript that addresses the points raised during the review process.

We look forward to receiving your revised manuscript.

Kind regards,

Imran Saeed, PhD

Academic Editor

PLOS ONE

2. Please amend your list of authors on the manuscript to ensure that each author is linked to an affiliation. Authors’ affiliations should reflect the institution where the work was done (if authors moved subsequently, you can also list the new affiliation stating “current affiliation:….” as necessary).

Additional Editor Comments:

Dear Authors,

I have carefully reviewed the reviewers' comments. Based on their feedback, I have decided that your manuscript requires major revisions. I trust the authors will address these points comprehensively. Additionally, please highlight all changes made and provide a detailed, point-by-point response to each reviewer’s comments.

Reviewers' comments:

Reviewer's Responses to Questions

**Comments to the Author**

1. Is the manuscript technically sound, and do the data support the conclusions?

Reviewer #1: Yes

Reviewer #2: Yes

2. Has the statistical analysis been performed appropriately and rigorously? 

Reviewer #1: Yes

Reviewer #2: Yes

3. Have the authors made all data underlying the findings in their manuscript fully available?

Reviewer #1: Yes

Reviewer #2: No

4. Is the manuscript presented in an intelligible fashion and written in standard English?

Reviewer #1: Yes

Reviewer #2: No

5. Review Comments to the Author

Reviewer #1: The topic, "A Qualitative Study about Chinese Teachers' Perceptions and Practices of Meritocracy," is interesting. I thoroughly read the manuscript and found some areas that need improvement to enhance the quality of the paper. Some of my suggestions are as follows:

1. The authors need to focus on improving the abstract. As the first part of the manuscript, it is crucial for captu1.ring the reader's attention. In my opinion, the abstract is too brief. Please consider making it more engaging and comprehensive.

2. The concept of meritocracy is not adequately explained. Since it is the central theme of the manuscript, the authors must provide a more detailed and thorough explanation of this concept.

3. Why are meritocracy and education important? What are the reasons behind this? The authors should focus on these questions and support their discussion with relevant and recent literature, as many studies have explored meritocracy.

4. Why focus only on teachers' perceptions and practices regarding meritocracy? The rationale behind this focus must be clarified, using the latest and relevant references.

5. The concept of meritocracy in the Chinese context must be clearly articulated and supported with appropriate references.

6. The methods section is insufficient and requires a more detailed explanation.

7. I agree with the analysis section; it is well-written and explained.

8. The discussion is good, but the limitations of the study and future directions are missing. These should be included.

Reviewer #2: Dear Editor,

Thank you for providing me with the opportunity to review this manuscript. I have read the complete manuscript and found it to be of good quality. However, there is always room for improvement. Below are some changes that need to be incorporated before publication:

1.The abstract is very brief and not well-written. It should be expanded and improved for clarity.

2. The introduction is overall well-written and logically coherent. However, it lacks the latest references, which could help strengthen the research gap.

3. In the literature review, the authors developed strong arguments on "Meritocracy and Education," "Teachers’ Perceptions of and Practices Regarding Meritocracy," and "Meritocracy in the Chinese Context." However, there are no recent references that support the literature with relevant theories.

4. The methods section should be clearer to the reader. On what basis were these school teachers from Shenzhen City selected? This section must be more understandable.

5. The results section is well-written and well-explained by the authors. The proposed questions are clearly addressed and interpreted effectively.

6.Limitations and future directions are missing and should be included.

6. PLOS authors have the option to publish the peer review history of their article (what does this mean? ). If published, this will include your full peer review and any attached files.

**Do you want your identity to be public for this peer review?** For information about this choice, including consent withdrawal, please see our Privacy Policy .

Reviewer #1: **Yes: ** Wang Jiatong

Reviewer #2: No

---

## [Author Response · Author response to Decision Letter 0]

16 Nov 2024

1.Publishing Editor：

A rebuttal letter that responds to each point raised by the academic editor and reviewer(s). A marked-up copy of your manuscript that highlights changes made to the original version.

respond:

We carefully considered all reviewer comments when we revised the whole manuscript. We will submit a list of responses along with revised manuscript.

reviewers

1.The authors need to focus on improving the abstract. As the first part of the manuscript, it is crucial for capturing the reader's attention. In my opinion, the abstract is too brief. Please consider making it more engaging and comprehensive.

respond:

As the reviewers suggested that we have rewrite for its clarity.

2.The introduction is overall well-written and logically coherent. However, it lacks the latest references, which could help strengthen the research gap.

respond:

Regarding the suggestion about the introduction, we introduced new literature on Chinese attitudes toward meritocracy. We analyzed the emerging ‘lying flat’ movement, the varying findings on people’s meritocratic beliefs in unequal environments, and changes in the labor market.

3.The authors developed strong arguments on "Meritocracy and Education," "Teachers’ Perceptions of and Practices Regarding Meritocracy," and "Meritocracy in the Chinese Context." However, there are no recent references that support the literature with relevant theories.

respond:

The discussion of meritocracy has produced many classic theories, meaning that some dated studies are inevitably referenced. However, we have still include the latest research to provide fresh insights. Such as, ideas about meritocracy from Platz (2020, 2022) and Mulligan (2018).

For the part of “Meritocracy and Education”, we included ideas about meritocracy as a society system from Bandiera (2024), Moreira (2022), Liu(2023) and Bruni (2021).

We added ideas about school meritocracy and teachers’ perceptions and practices from recent references,such as Castillo et al. (2022), Gonçalves et al.(2024), Krumer (2022) and Doyle et al. (2023).

Regarding “Meritocracy in the Chinese Context,” we have added the typical features of meritocracy in China. We would like to clarify that we have included recent research and official data to support the arguments, such as the employment of college graduates in 2022 and the number of graduate students enrolled in 2023.

4.The concept of meritocracy is not adequately explained. Since it is the central theme of the manuscript, the authors must provide a more detailed and thorough explanation of this concept.

respond:

Regarding the suggestion about the literature, we restructured this section. We have thoroughly explained the concept of meritocracy from both the philosophical perspective of justice theory and as a social model. Using the theory of desert, we have clarified what merit means in the context of distributive justice. As a social model, we discussed the potential political and economic effects of a society sorted around merit.

5.Why are meritocracy and education important? What are the reasons behind this? The authors should focus on these questions and support their discussion with relevant and recent literature, as many studies have explored meritocracy.

respond:

We utilized Bell’s classic theory, education-based meritocracy, to discuss the role of education in meritocracy. Our discussion is also combined with the arguments of lately studies to support the view of this paper that education is the cornerstone of meritocracy.

6.Why focus only on teachers' perceptions and practices regarding meritocracy? The rationale behind this focus must be clarified, using the latest and relevant references.

respond:

We have rephrased this section to explain why we focus on teachers. After briefly outlining the foundational role of schools in promoting meritocracy, we then analyzed the influence of teachers’ behaviors and views in this process.

Example:

Schools’ contribution to meritocracy is closely linked to teachers (Castillo et al, 2022; Gonçalves et al., 2024), making their practices and beliefs key to understanding meritocracy and social inequity.

An international study by Batruch et al. (2023) found that teachers with meritocratic beliefs, rather than perceiving unfairness and supportive of affirmative action policies, encourage students to work hard, thereby perpetuating and strengthening meritocratic beliefs within schools.

7.The concept of meritocracy in the Chinese context must be clearly articulated and supported with appropriate references.

respond:

Based on existing research, we summarized the characteristics of meritocracy in China, highlighting its features as a governance approach.

8.The methods section is insufficient and requires a more detailed explanation.

respond:

Regarding the suggestion on how to invite participants, we have added a brief description outlining the criteria followed for inviting participants in our study.

We have detailed the coding process and identified themes on the basis of the concept of meritocracy and teachers’ perceptions and teaching practices. The data were thematically coded according to meritocratic and non-meritocratic factors within the concept of meritocracy. The secondary coding refined the following themes: views of meritocratic factors and guanxi, and teaching practices, considering that guanxi were frequently mentioned.

9.The limitations of the study also need to be discussed in this section.

respond:

We have added a new paragraph to discuss the limitations of the study.

---

## [Decision Letter · Decision Letter 1]

7 Feb 2025

PONE-D-24-28202R1A Qualitative Study of Chinese Teacher’s Perceptions and Practices of Meritocracy

PLOS ONE

Dear Dr. Sun,

Thank you for submitting your manuscript to PLOS ONE. After careful consideration, we feel that it has merit but does not fully meet PLOS ONE’s publication criteria as it currently stands. Therefore, we invite you to submit a revised version of the manuscript that addresses the points raised during the review process.

Below, I have outlined the revisions/improvements needed:

**Methodology:**Include a table summarizing the demographic characteristics of teachers in each school to clarify differences.Clearly state whether patterns of responses were similar or varied across schools.Provide the full interview questions in the text or appendix (both in Mandarin and English).Clarify if interviews were conducted separately or together and describe how consistency was ensured across interviewers.Elaborate on the coding process—who conducted independent coding and how consensus was reached.
**Conceptual Framework, Discussion, and Conclusion:**Consider adding a conceptual diagram to illustrate how privilege (e.g., home resources, social networks) and merit (e.g., effort, talent) contribute to student achievement.Clarify the causal relationship between student background advantages and effort in determining success.Highlight how teachers emphasize effort as the key factor influencing student outcomes, even though structural constraints exist.Expand the discussion on structural constraints and their impact on teachers’ agency.Provide a more in-depth exploration of the implications for policy and practice.
**Language & Clarity:**Clarify whether "150 RMB" refers to daily per capita income.Provide more context for the term *ren qing shi gu* to ensure non-Chinese readers understand its full meaning.Justify or cite the claim that “students attribute academic success to three parts talent and seven parts effort.”

By addressing these comments, you can improve the comprehensiveness, relevance, and clarity of your research article.

We look forward to receiving your revised manuscript.

Kind regards,

Mc Rollyn Daquiado Vallespin

Academic Editor

PLOS ONE

Journal Requirements:

Reviewers' comments:

Reviewer's Responses to Questions

**Comments to the Author**

1. If the authors have adequately addressed your comments raised in a previous round of review and you feel that this manuscript is now acceptable for publication, you may indicate that here to bypass the “Comments to the Author” section, enter your conflict of interest statement in the “Confidential to Editor” section, and submit your "Accept" recommendation.

Reviewer #1: All comments have been addressed

Reviewer #2: All comments have been addressed

Reviewer #3: All comments have been addressed

Reviewer #4: (No Response)

Reviewer #5: All comments have been addressed

2. Is the manuscript technically sound, and do the data support the conclusions?

Reviewer #1: Yes

Reviewer #2: Yes

Reviewer #3: Yes

Reviewer #4: Yes

Reviewer #5: Partly

3. Has the statistical analysis been performed appropriately and rigorously? 

Reviewer #1: N/A

Reviewer #2: N/A

Reviewer #3: Yes

Reviewer #4: Yes

Reviewer #5: N/A

4. Have the authors made all data underlying the findings in their manuscript fully available?

Reviewer #1: Yes

Reviewer #2: Yes

Reviewer #3: Yes

Reviewer #4: Yes

Reviewer #5: No

5. Is the manuscript presented in an intelligible fashion and written in standard English?

Reviewer #1: Yes

Reviewer #2: Yes

Reviewer #3: Yes

Reviewer #4: Yes

Reviewer #5: Yes

6. Review Comments to the Author

Reviewer #1: The draft has been improved, and all my suggestions have been incorporated. From my side, it is now acceptable.

Reviewer #2: I have thoroughly reviewed the revise draft, and it has been improved. It is now acceptable to me. good luck.

Reviewer #3: Improve your technical writing but its ok after reviewed the manuscript from the first, second reviwers.

Reviewer #4: Discussion and conclusions

1. Lack of depth in the discussion of structural constraints and their impact on teachers' agency

2. Limited exploration of the implications for policy and practice

Reviewer #5: Please see the attached file.

7. PLOS authors have the option to publish the peer review history of their article (what does this mean? ). If published, this will include your full peer review and any attached files.

**Do you want your identity to be public for this peer review?** For information about this choice, including consent withdrawal, please see our Privacy Policy .

Reviewer #1: No

Reviewer #2: No

Reviewer #3: **Yes: ** Dr. IDRIS ADAMU CURRICULUM & INSTRUCTION AMINU SALEH COLLEGE OF EDUCATION AZARE. BAUCHI STATE NIGERIA

Reviewer #4: No

Reviewer #5: **Yes: ** Nathaniel D. Porter

---

## [Author Response · Author response to Decision Letter 1]

4 Mar 2025

We have addressed all the comments, and the details can be found in the document titled “Response to Reviewers(minor revised）”.

---

## [Editor Report · Decision Letter 2]

7 Mar 2025

A Qualitative Study of Chinese Teacher’s Perceptions and Practices of Meritocracy

PONE-D-24-28202R2

Dear Dr. SUN,

We’re pleased to inform you that your manuscript has been judged scientifically suitable for publication and will be formally accepted for publication once it meets all outstanding technical requirements.

Kind regards,

Mc Rollyn Daquiado Vallespin

Academic Editor

PLOS ONE
---

## [Editor Report · Acceptance letter]

PONE-D-24-28202R2

PLOS ONE

Dear Dr. Sun,

I'm pleased to inform you that your manuscript has been deemed suitable for publication in PLOS ONE. Congratulations! Your manuscript is now being handed over to our production team.

Kind regards,

on behalf of

Dr. Mc Rollyn Daquiado Vallespin

Academic Editor

PLOS ONE